# Narrative–affect discrepancy as a regulated degree of freedom in 351,734 relationship narratives

**Ryan SangBaek Kim** [ID]*

Ryan Research Institute (RRI), Paris, France

* ryan@ryanresearch.org

## Abstract

In naturalistic emotional narratives, the intensity of expressed affect does not scale proportionally with narrative structure. Using 351,734 English-language relationship narratives from online support communities (the ANEST Narrative–Affect Dataset, ANAD v1.1.0), we constructed a two-dimensional expressive space defined by narrative complexity ($N$) and linguistically inferred affective intensity ($A$), with their signed discrepancy ($D = N - A$) treated as a derived coordinate. Rather than converging toward low discrepancy, human narratives occupied a broad but structured space consistent with trade-offs between relational exposure and cognitive effort. We identified four empirically separable regimes of expressive organization: coupled expression (non-extreme discrepancy; the complement of the extreme regimes), strategic understatement (high $A$, low $N$, $D < 0$), strategic overstatement (high $N$, low $A$, $D > 0$), and collapse (high $A$ with limited narrative scaffolding). A data-anchored cost model (NCS) formalized these regimes as arising from the interaction of exposure risk and cognitive effort, not from discrepancy minimization per se. Coupled expression dominated the corpus (91.3%), while the remaining regimes formed smaller but non-negligible subpopulations (Understatement: $n = 20,223$; Collapse: $n = 8,040$; Overstatement: $n = 2,223$), indicating that extreme discrepancy configurations occur systematically rather than as isolated outliers. As a comparative probe, we projected an RLHF-aligned large language model into the same space using matched prompts and identical feature extraction. Because $D$ is a deterministic function of $N$ and $A$, expressive extent was quantified as convex hull area in the clipped ($N', A'$) plane. Under this procedure, the model occupied a markedly smaller region (approximately $1.70\times$ smaller hull area; bootstrap 95% CI [1.68, 1.70]; permutation $p < 0.0001$) and concentrated near low-discrepancy configurations, with sparse occupancy of extreme under- and overstatement regimes. Together, these findings suggest that narrative–affect discrepancy is a measurable and regulated dimension of emotional expression and provide a reproducible geometric basis for comparing expressive degrees of freedom across populations and systems.

provided the original author and source are credited.

**Data availability statement:** The data underlying the results presented in this study are available from Zenodo. The ANEST Narrative-Affect Dataset (ANAD) is available at https://doi.org/10.5281/zenodo.17632585 (all-versions DOI). Analyses use derived features from v1.1.0 (DOI: 10.5281/zenodo.17981111); the canonical feature table in v1.3.0 (DOI: 10.5281/zenodo.18680687) can also be used for reproduction. All derived data files, analysis outputs, and reproducible code are provided within the Supporting information and Supplementary Software.

**Funding:** The author(s) received no specific funding for this work.

**Competing interests:** The author has declared that no competing interests exist.

## Introduction

The dominant picture in psychology, psychiatry, and affective computing assumes that healthy communication is fundamentally *coherent*: what is experienced is faithfully encoded into language, and persistent gaps between what is experienced and what is expressed signal pathology, defensiveness, or noise [1–15]. Therapeutic models often treat greater congruence between verbal report and lived experience as a marker of progress, while algorithms for sentiment analysis and emotion recognition implicitly assume that textual surface valence can be treated as a reliable proxy for underlying affect [16–18].

Yet everyday relational life suggests a more complex picture. People routinely downplay pain to protect autonomy, exaggerate disappointment to signal boundaries, or wrap intense affect in minimal words to avoid burdening others [19–21]. In these cases, divergence between what is lived and what is narrated need not be a failure of representation; it can be an adaptive strategy under social constraints. The gap itself—the *discrepancy* between narrative structure and expressed affect—may carry structure and purpose.

Classical emotion theories gesture toward this tension. Work on emotion regulation highlights reappraisal, suppression, and expressive modulation as active processes shaping the trajectory from experience to expression [1,22,23]. Social constructivist accounts emphasize that emotional episodes are co-authored with audiences and cultural scripts [6,24–30]. Yet despite these insights, empirical operationalizations rarely center the discrepancy itself—the magnitude and sign of the mismatch between narrative structure and expressed affect—as a primary object of measurement. Divergence is typically relegated to a residual term: random error, bias, or unmodelled variance.

As a secondary comparative analysis, we later project an RLHF-aligned large language model into the same coordinate system using matched prompts and an identical scoring pipeline, under a single prompt and decoding configuration, not as a benchmark of emotional understanding but as a geometric contrast under the same measurement procedure.

We therefore ask a simpler empirical question: across large samples of naturalistic narratives, is narrative–affect discrepancy organized in systematic, measurable ways? If so, what patterns dominate the joint space of narrative structure and expressed affect, and can these patterns be characterized using transparent, reproducible methods?

To address this question, we shift the analytic focus away from attempting to directly measure subjective feeling—which is inherently private, context-dependent, and only partially reportable—and instead examine the *structural degrees of freedom* available within language itself. Narratives are not passive containers of affect. Their length, lexical diversity, syntactic articulation, and semantic density reflect active construction choices that can be modulated independently of affective intensity [31–33]. These choices impose costs: longer narratives demand cognitive effort, while sparse expression of intense affect risks misunderstanding or relational rupture. Table 1 summarizes these constructs and the four expressive regimes derived from them.

**Table 1. Overview of key constructs and expressive regimes.**

| Construct | Definition | Role |
|---|---|---|
| N | Narrative complexity (LoC, 0–10) | Structural axis |
| A | Affective intensity (sentiment_norm, 0–10) | Affective axis |
| D = N − A | Signed discrepancy | Derived coordinate |
| Regime | Condition | Interpretation |
| Coupled | \|D\| below extreme thresholds | Modal expressive pattern |
| Understatement | High A, low N, D < 0 | Compressed scaffolding |
| Overstatement | High N, low A, D > 0 | Elaboration without affect |
| Collapse | High A, very low N, large \|D\| | Structural insufficiency |

We therefore conceptualize emotional expression as operating within a constrained state space defined by three empirically accessible axes:

1. **Narrative complexity ($N$)**: a structural measure of how much linguistic scaffolding is deployed, operationalized via length, lexical diversity, and semantic elaboration metrics (see Methods).

2. **Linguistically inferred affective intensity ($A$)**: the intensity of affect expressed in language, estimated using continuous sentiment-derived measures designed to capture magnitude beyond polarity [16–18].

3. **Narrative–affect discrepancy ($D = N − A$)**: the signed difference between structural elaboration and expressed affective intensity.

Importantly, $D$ is not introduced as an estimate of hidden "true" feeling. Rather, it indexes how much narrative structure is deployed relative to the affective signal expressed in language. When $A$ is high and $N$ is low, expression is compressed; when $N$ is high and $A$ is low, expression is expansive. In both cases, discrepancy reflects a strategic allocation of expressive resources rather than a failure of reporting.

A clarification on terminology is warranted. Throughout this paper, the term "regulated" is used in a descriptive, population-level sense. It denotes persistent distributional constraint patterns observable in aggregate data, not conscious psychological control or homeostatic optimization by individuals. The usage is analogous to how constrained degrees of freedom are described in physical systems: the constraint is a property of the observed distribution, not an attributed intention of any particular agent. This distinction is maintained throughout the analyses that follow.

This framing departs from alignment-based assumptions common in both psychology and affective computing, where $D \approx 0$ is often treated as an implicit ideal. Instead, we ask whether humans systematically occupy non-zero regions of discrepancy space, and whether these regions exhibit stable regimes indicative of constrained regulation.

Using 351,734 English-language relationship narratives collected from online advice forums and support communities, we construct the *Narrative–Control Space* (NCS). We analyze the joint distribution of ($N$, $A$, $D$) and test three central claims:

1. **Discrepancy is structured, not residual.** Human narratives occupy a broad but non-uniform space in which narrative complexity ($N$) and expressed affective intensity ($A$) are only weakly coupled, yielding stable and frequently visited non-zero discrepancy configurations.

2. **Robust regimes of organization emerge.** The empirical distribution supports four reproducible regimes—coupled expression, strategic understatement, strategic overstatement, and collapse—capturing distinct configurations of narrative scaffolding relative to expressed affect.

3. **A secondary geometric comparison is feasible under identical measurement.** When an RLHF-aligned language model is scored using matched prompts and the same feature extraction pipeline, its occupancy concentrates within a smaller region of the human-derived coordinate system, enabling a contrastive analysis of accessible expressive space under a shared procedure.

Finally, we use the human-derived NCS as a comparative geometry to analyze an aligned language model. This comparison is not intended as a benchmark of emotional "understanding," but as a contrastive probe: if human communication relies on access to multiple discrepancy regimes, what expressive regions remain accessible under alignment constraints?

This work contributes to three literatures. First, it extends emotion regulation research by identifying narrative structure itself as an independent expressive degree of freedom, quantifiable at scale. Second, it complements narrative psychology by introducing a geometric, population-level characterization of structure–affect relationships. Third, it offers a descriptive lens on alignment in language models, framing its effects in terms of expressive area contraction rather than accuracy or safety alone.

We do not claim to measure subjective feeling directly, nor do we assume that linguistic affect exhausts emotional experience. Instead, we focus on what can be robustly observed: how much narrative structure is deployed relative to expressed affect, and how this balance is systematically organized across hundreds of thousands of real-world narratives.

## Materials and methods

### Ethics statement

This study analyzes the ANEST Narrative–Affect Dataset (ANAD v1.1.0), which was derived from publicly accessible online narratives and processed to remove or mask personally identifying information (PII) prior to release. The Zenodo v1.1.0 release contains derived features only and explicitly excludes raw post text and user-level metadata, thereby reducing traceability and re-identification risk. The present work involves no interaction or intervention with human participants and analyzes de-identified text records. Ethical considerations therefore focus on privacy preservation and minimization of re-identification risk. We report the dataset provenance, filtering, and de-identification procedures as provided in the dataset release [34]. We do not reproduce verbatim narrative text in either the main text or the Supporting Information. All reporting is based on released derived features and aggregate summaries, consistent with the dataset's privacy-preserving design.

### Human dataset: ANAD v1.1.0 (ANEST Narrative–Affect Dataset)

Human narratives were drawn from the ANEST Narrative–Affect Dataset (ANAD v1.1.0), a curated collection of 351,734 English-language relationship narratives collected from publicly accessible online forums (2012–2023) and released as derived features only [34]. The release excludes raw post text and user-level metadata and is processed to reduce re-identification risk (see Ethics statement and the dataset record).

This study uses the dataset-provided precomputed variables: narrative complexity (`LoC`) and linguistically inferred affective intensity (`sentiment_norm`), both scaled to a common 0–10 range in the dataset pipeline [34]. The dataset also provides an absolute discrepancy feature (`NADI_abs`); however, for regime and geometric analyses we define signed discrepancy as

$$D = N - A, \quad \text{where } N = \texttt{LoC}, \ A = \texttt{sentiment\_norm}.$$

Throughout, $N$ and $A$ are treated as fixed released measurements, and all downstream steps (clipping, regime labeling, density estimation, and geometric occupancy) are applied identically to humans and the language model.

We operationalize signed discrepancy for analysis as

$$D = N - A,$$

which distinguishes *understatement* states ($D < 0$; expressed affect exceeds narrative structure) from *overstatement* states ($D > 0$; narrative structure exceeds affect). For descriptive reporting, we summarize discrepancy magnitude using $|D|$; regime and comparative analyses use signed $D$. Because $D$ is a deterministic linear function of $N$ and $A$, the triplet ($N$, $A$, $D$) lies on a plane in three-dimensional space. The effective dimensionality of the expressive geometry is therefore two. All geometric measures (convex hull area, occupancy entropy) are computed in the clipped ($N'$, $A'$) plane, with $D$ treated as a derived coordinate for regime labeling and interpretive purposes.

## LLM trajectories

Model trajectories consist of 1,000 single-turn outputs generated by an RLHF-aligned large language model (OpenAI `gpt-4.1-mini`; temperature 0.9, top-p 1.0, max tokens 600) [35–37] under a fixed decoding configuration. Outputs were scored using the same feature extractors and scaling pipeline applied to the human corpus, producing $N$, $A$, and $D$ on the same scale.

To support replication despite provider versioning, the provider/model identifier (or closest available tag), generation date range, the fixed prompt template (`llm_prompt_set.txt`), and decoding parameters (including any nucleus sampling, maximum tokens, and stopping criteria) are documented in Supporting Information (S7 Text) and the released configuration file `llm_config.json`. LLM trajectories are stored as `nadi_llm_trajectories.parquet`, and we restrict analysis to the first turn ($t = 0$) for comparability with single-narrative human records.

## Preprocessing and clipping

To reduce the influence of extreme outliers while preserving distributional structure, we compute the 1st and 99th percentiles for each primary dimension ($N$, $A$) from the human corpus and apply axis-wise clipping to both humans and the LLM:

$$N' = \mathrm{clip}(N; N_{1\%}, N_{99\%}), \quad A' = \mathrm{clip}(A; A_{1\%}, A_{99\%}).$$

Signed discrepancy in the clipped space is then computed as $D' = N' - A'$. All density maps, regime assignments, and geometric occupancy measures are computed on the clipped variables.

Crucially, the same human-derived clipping bounds are applied to both humans and the LLM; area differences therefore reflect comparative occupancy under a shared bounded geometry rather than agent-specific preprocessing.

## Density estimation and correlation analysis

Two-dimensional density maps are provided in Supporting Information (Fig S1 in S2 Text) and are computed using histogram-based estimators on the clipped human data. For each pair $(X, Y) \in \{(N', A'), (N', D'), (A', D')\}$, we construct a $60 \times 60$ grid, count observations per cell, and export triplets of cell centers and counts as: `fig1_density_NA.dat`, `fig1_density_ND.dat`, and `fig1_density_AD.dat`. Visualization uses logarithmic scaling ($\ln(\rho + 1)$) to preserve dynamic range.

Pearson correlations (S4 Table in S4 Text) are computed over the clipped human data and exported to `table2_corr_matrix.dat`. Because $D$ is defined as $N - A$ prior to clipping, the $A$–$D$ association is structurally constrained; we therefore interpret correlations primarily in terms of remaining degrees of freedom, with particular emphasis on the empirical near-independence of $N$ from $A$ and $D$.

## Regime definitions and group comparisons

We operationalize discrepancy regimes in two complementary ways.

**Four-regime labeling (Fig 1).** We define four expressive regimes using empirical quantile thresholds estimated from the full human corpus on the bounded 0–10 scale. Let $\tau$ denote the 75th percentile of $|D|$, $A_{\text{high}}$ the 75th percentile of $A$, and $N_{\text{low}}$ the 25th percentile of $N$. We assign: (i) *Collapse* if $A \geq A_{\text{high}}$, $N \leq N_{\text{low}}$, and $|D| \geq \tau$; (ii) *Understatement* if $A \geq A_{\text{high}}$ and $D \leq -\tau$ (excluding Collapse); (iii) *Overstatement* if $D \geq \tau$ (excluding Collapse); and (iv) *Coupled* otherwise, i.e., the complement of extreme discrepancy regimes. Regime labels are computed for the full corpus and exported as `fig2_regimes.dat`. Fig 1 visualizes a stratified subset (`fig2_regimes_sampled.dat`) with a fixed point cap for LaTeX compilation stability; all prevalence and summary statistics are computed on the full corpus.

Regime boundaries are quantile-based and data-driven. Sensitivity analyses (S1 Text) confirm that qualitative regime structure and contraction findings are stable under $\pm 5$ percentile threshold shifts. Coupled expression is defined as the complement of the three extreme regimes and is not posited as a theoretically privileged equilibrium.

We use the term "collapse" rather than "compression" because the pattern involves not merely reduced expressive range but a convergence of narratives toward the lower structural boundary of the space while affective intensity remains high, a qualitative shift distinct from proportional scaling.

**Signed-quartile bands (Table 2).** To compare humans and the LLM with a single, shared partition, we compute the 25th and 75th percentiles $(Q_1, Q_3)$ of the signed human discrepancy $D$ and define: *Understatement* $(D < Q_1)$, *Intermediate* $(Q_1 \leq D \leq Q_3)$, and *Overstatement* $(D > Q_3)$. The same cutoffs are applied to the LLM to compute group-conditional means. These results are exported as `table3_group_compare.dat`.

## Data-driven cost landscape

To formalize expressive regulation as constrained control, we construct a cost function $J(N', A', D')$ using only statistics estimated from the clipped human data. We compute min–max parameters $(N_{\text{min}}, N_{\text{max}})$, $(A_{\text{min}}, A_{\text{max}})$, $(D_{\text{min}}, D_{\text{max}})$ and define normalized variables

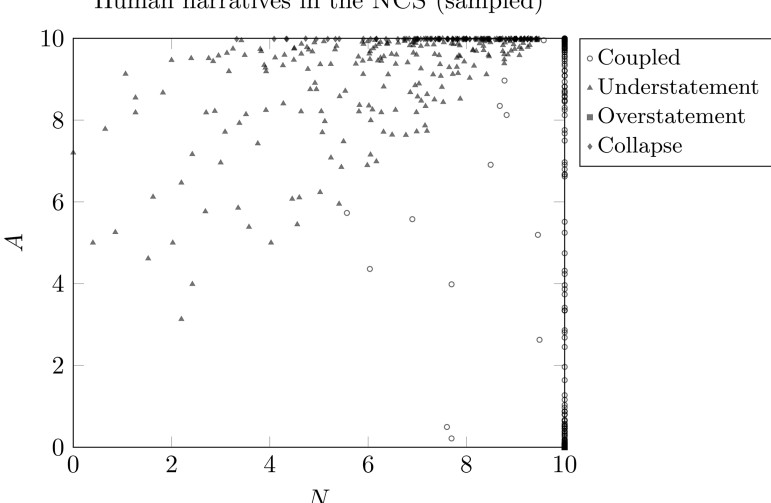

Human narratives in the NCS (sampled)

**Fig 1. A stratified visualization sample of human narratives in the NCS (*n* = 200 for readability).** Regime definitions and prevalence are computed on the full corpus and reported in Table 3 and Methods.

**Table 2. Group-conditional means for humans and the aligned model (shared human cutoffs).**

| Group | N | A | D | Agent |
|---|---|---|---|---|
| Understatement | 1.1742169077188636 | 9.712766137102827 | −8.538549229383962 | Human |
| Understatement | 0.5513932550436986 | 8.644823993409632 | −8.093430738365935 | LLM |
| Intermediate | 2.7838726975989077 | 7.908712286627319 | −5.124839589028411 | Human |
| Intermediate | 1.4010661596515022 | 4.424860624572343 | −3.0237944649208415 | LLM |
| Overstatement | 3.013617509185932 | 0.456845440898856 | 2.5567720682870756 | Human |
| Overstatement | 3.2775034695174976 | 1.8604788662996494 | 1.4170246032178482 | LLM |

$$N_n = \frac{N' - N_{\min}}{N_{\max} - N_{\min} + \varepsilon}, \quad A_n = \frac{A' - A_{\min}}{A_{\max} - A_{\min} + \varepsilon}, \quad D_n = \frac{D' - D_{\min}}{D_{\max} - D_{\min} + \varepsilon},$$

with a small $\varepsilon$ to avoid division by zero.

Let $D^*$ be the median of the *signed* human discrepancy distribution $D'$ computed on the clipped human data (and $D_n^*$ its normalized value). We define three components:

$$\text{Risk}(A', D') = A_n^2 \left(D_n - D_n^*\right)^2, \tag{1}$$

$$\text{Effort}(N') = N_n^2, \tag{2}$$

$$\text{Deviation}(D') = \left(D' - D^*\right)^2. \tag{3}$$

Weights are fixed from empirical dispersion using inverse-variance style scaling:

$$\lambda_N = \frac{1}{\text{SD}(N')}, \quad \lambda_A = \frac{1}{\text{SD}(A')}, \quad \lambda_D = \frac{1}{\text{SD}(|D'|)},$$

computed on the clipped human sample. We use $\text{SD}(|D'|)$ to scale deviation by typical discrepancy magnitude irrespective of sign. The composite cost is

$$J(N', A', D') = \lambda_A \cdot \text{Risk}(A', D') + \lambda_N \cdot \text{Effort}(N') + \lambda_D \cdot \text{Deviation}(D'). \tag{4}$$

Several design decisions warrant explanation. Quadratic penalties are adopted for parsimony and analytic tractability. The deviation component is centered on the empirical median $D^*$ rather than on zero because the population does not converge toward zero discrepancy; centering on the observed median acknowledges this non-zero anchor as an empirical regularity rather than a normative target. The deviation term penalizes departure from typical discrepancy magnitude, not departure from zero; the cost landscape thus describes bounded variation around an empirical attractor, distinct from a model in which discrepancy is globally minimized. Taken together, $J$ is an abductive, illustrative function. It was not fitted to individual trajectories nor inferred as a latent objective function. Its purpose is to provide a parsimonious geometric summary of the constraint structure consistent with the observed multi-regime occupancy.

We evaluate $J$ on a regular $60 \times 60 \times 60$ grid spanning empirical ranges and export the full landscape as `fig3_cost_landscape.dat`, along with slices for visualization (e.g., `fig3_cost_slice.dat`). Qualitatively similar basin structure

and contraction patterns were obtained under alternative weighting perturbations (Supporting Information (S1 Text)), supporting robustness to reasonable rescalings.

### Expressive area and occupancy entropy

We quantify occupancy in the clipped $(N', A')$ plane using two complementary measures:

**Convex hull area.** We estimate the two-dimensional convex hull area for each agent. Because $D = N - A$ is a deterministic function of the two primary axes, the triplet $(N, A, D)$ is confined to a plane in three-dimensional space and its convex hull volume is structurally zero. We therefore report hull area in the $(N', A')$ plane, which captures the effective expressive extent without dependence on clipping artifacts. To control for sample size differences, we compute the human area via bootstrap resampling: 100 random subsets of 1,000 human trajectories (matching the LLM count) are drawn and their hull areas averaged. The resulting mean human area is compared against the single-hull estimate for the LLM. These results are exported to `fig4_volume_compare.dat`.

**Occupancy entropy.** We compute a two-dimensional histogram over $(N', A')$ using $24 \times 24$ bins, normalize counts to probabilities, and compute Shannon entropy $H = -\sum_i p_i \log p_i$. Entropy is interpreted as uniformity of occupancy conditional on the explored region, complementary to convex area.

### Software

All preprocessing and analyses were conducted in Python 3.12 using standard scientific libraries (e.g., `numpy`, `pandas`, `scipy`). As a methodological precedent for making high-inference constructs auditable via explicit, theory-grounded operationalization, we follow the same reproducibility-first design philosophy articulated in DefMoN [38], while applying it here to narrative–affect discrepancy geometry rather than defensiveness-oriented synthesis. A reproducible software bundle (Supplementary Software) documents the required dependencies and scripts used to generate the exported Source Data files. Figures are rendered in LaTeX via `pgfplots` and `pgfplotstable`, reading directly from exported `.dat`/`.csv` source files to ensure traceability between reported quantities and analysis outputs. No hand-tuned or simulated values were introduced into tables or figures.

## Results

### Global geometry of narrative–affect discrepancy

We first examined the joint distribution of narrative complexity ($N$) and linguistically inferred affective intensity ($A$) across the full NCS corpus ($n = 351{,}734$). Supporting Information (Fig S1 in S2 Text) shows density projections in the $(N,A)$, $(N,D)$, and $(A,D)$ planes, and S4 Table in S4 Text reports the corresponding correlation structure. Affective intensity spans a wide range, while narrative complexity varies substantially even within narrow affect bands.

The absence of strong diagonal structure in the $(N,A)$ projection suggests that discrepancy does not arise from a simple coupling between affective intensity and narrative complexity. Instead, the projections reveal broad near-independence between $N$ and $A$ ($r = 0.009$; S4 Table in S4 Text), confirming that narrative complexity and affective intensity vary as largely independent dimensions. Fig S1 (Supporting Information, S2 Text) is read as a map of *allowed configurations* in the empirical geometry, motivating the regime-based and cost-based analyses that follow.

Fig S1 (Supporting Information, S2 Text) also reveals that the marginal density of $D$ is not concentrated narrowly around a single value. Instead of converging toward low-discrepancy states, human narratives occupy a wide expressive range, suggesting that non-zero discrepancy is a stable and frequently visited configuration.

### Regime identification in discrepancy space

To characterize this structure, we partitioned the $(N,A,D)$ space into regions defined by the sign and magnitude of discrepancy and by relative narrative elaboration. Using empirical quantile thresholds computed from the human corpus (see Methods), we identify four robust regimes of discrepancy organization:

1. **Coupled expression**: the complement of the extreme discrepancy regimes, capturing the dense majority of narratives that do not meet the thresholds for strong under- or overstatement or collapse.

2. **Strategic understatement**: high $A$ paired with relatively sparse narrative structure ($A \gg N$, $D < 0$).

3. **Strategic overstatement**: low $A$ accompanied by extensive narrative elaboration ($N \gg A$, $D > 0$).

4. **Collapse**: high $A$ with comparatively low narrative scaffolding *and* large-magnitude discrepancy, corresponding to a higher-cost region in which affective magnitude is expressed with minimal structural containment.

Throughout, the collapse regime is defined descriptively in terms of structural insufficiency relative to affective magnitude, without implying clinical pathology or dysfunction.

Table 3 reports regime-level summary statistics, including prevalence and mean values of ($N,A,D$). Coupled expression accounts for the majority of narratives (91.3%), while Understatement ($n = 20{,}223$), Collapse ($n = 8{,}040$), and Overstatement ($n = 2{,}223$) remain clearly represented. Thus, extreme discrepancy configurations are not isolated artifacts but systematic, measurable modes within the population distribution.

## Cost landscape in narrative–control space

To interpret these regimes mechanistically, we define a data-anchored cost function $J(N,A,D)$ that trades off exposure risk, cognitive effort, and deviation from a typical discrepancy target (Methods). This construction does not assume that individuals consciously optimize $J$; rather, it provides a parsimonious quantitative description of constraint structure consistent with multi-regime occupancy.

Fig 2 visualizes an exported slice of the cost landscape over ($N,A$) with point color indicating cost. Importantly, low-cost basins are not confined to $D = 0$ states; rather, distinct regions correspond to understatement and overstatement configurations under different trade-offs.

This structure implies that discrepancy is not penalized per se. Rather, costs arise when expressive strategies fail to balance narrative scaffolding against affective intensity. Collapse corresponds to a higher-cost region in which affective magnitude is high but narrative resources are insufficient to regulate exposure.

## Expressive area and regime occupancy

We next quantified the *expressive area* accessible to human narrators by computing the convex hull of occupied regions in the ($N,A$) plane. To ensure robust comparison and reduce tail leverage, we perform axis-wise clipping to human-derived percentile bounds and compute geometry in the clipped space (Methods). This area captures the range of discrepancy strategies empirically realized across the population under bounded observation.

Human narratives occupy a large and topologically rich area spanning all four regimes. Regime occupancy is continuous rather than discrete, with smooth transitions between modes. This suggests that expressive regulation operates along gradients rather than fixed categories.

Area captures the reachable extent of expression in the clipped space, complementing regime-based summaries of where narratives concentrate.

**Table 3. Regime-level summary statistics and prevalence in the NCS (rendered from exported CSV).**

| Regime | $\overline{N}$ | $\overline{A}$ | $\overline{D}$ | Count | Prevalence (%) |
|---|---|---|---|---|---|
| Coupled | 2.4779036488616297 | 6.235895216468274 | −3.7579915676066435 | 321248 | 9.1333e1 |
| Understatement | 1.677783667588434 | 9.960305147604213 | −8.28252148001578 | 20223 | 5.7495e0 |
| Overstatement | 9.414798910252461 | 0.1864302744039586 | 9.228368635848502 | 2223 | 6.3202e-1 |
| Collapse | 0.8658058427117803 | 9.95244620646766 | −9.086640363755881 | 8040 | 2.28583e0 |

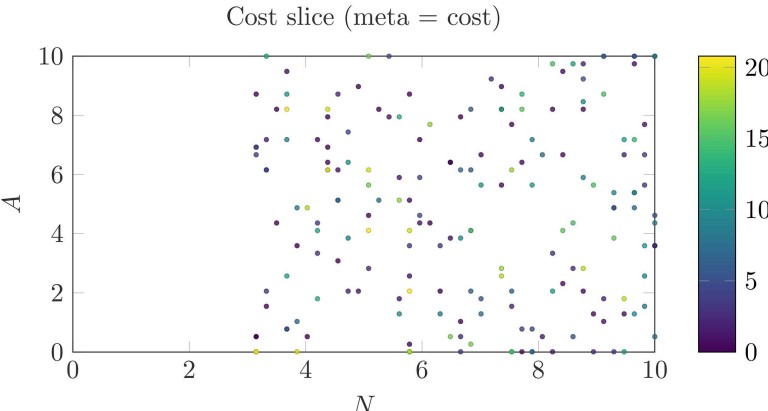

**Fig 2. Data-anchored cost over (N, A) (slice at D\* in the clipped space; Methods).** Low-cost basins correspond to configurations that trade off exposure risk (high *A*) against narrative effort (high *N*), rather than globally minimizing discrepancy. Understatement-leaning basins appear at higher *A* with lower *N*, whereas overstatement-leaning basins appear at higher *N* with lower *A*.

## Comparison with an aligned language model

Finally, we projected an RLHF-aligned large language model into the same NCS using matched prompts and identical feature extraction procedures (Methods). The model's expressive area is markedly contracted, occupying a narrow band concentrated near low-discrepancy states. Quantitatively, the model hull area is approximately 1.70 times smaller than the human hull area (human bootstrap mean: 99.51; LLM: 58.68; bootstrap 95% CI for contraction ratio: [1.68, 1.70]; permutation $p < 0.0001$, T = 10,000 shuffles; Fig 3; S6 Text). Notably, regions corresponding to strategic understatement and overstatement are sparsely populated, and the collapse regime is almost entirely absent.

Occupancy entropy, which measures uniformity of distribution within the explored region, was 4.654 for humans and 5.103 for the LLM. The higher LLM entropy indicates that the model distributes more uniformly within its narrower occupied region, whereas humans concentrate in specific regime zones despite spanning a broader area.

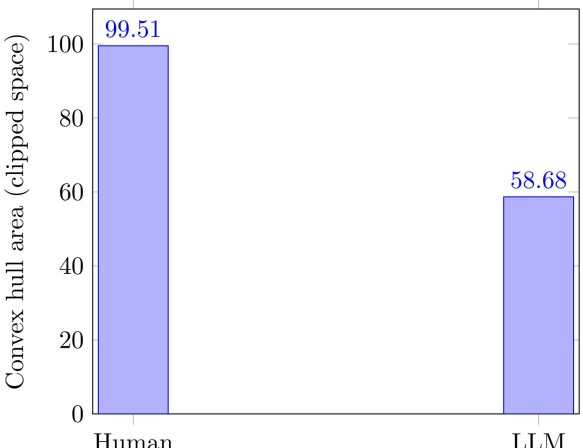

**Fig 3. Convex hull area for humans and the LLM in the clipped (N′, A′) plane (Methods).** The human value denotes the bootstrap-matched mean (100 resamples, *n* = 1,000 each). Contraction ratio: 1.70× (95% CI [1.68, 1.70]; permutation *p* < 0.0001). Full uncertainty reporting in S6 Text.

This contraction is not attributable to reduced affective range alone. Rather, it reflects a joint restriction on how narrative structure and affective intensity are combined. The model produces contextually coherent responses, but with limited variation in discrepancy strategies.

Taken together, these results show that humans systematically organize emotional expression across multiple discrepancy regimes, while alignment-constrained language models inhabit a much narrower expressive geometry.

## Discussion

### Narrative–affect discrepancy as a measurable regulatory dimension

The present findings challenge the assumption that effective emotional expression necessarily minimizes discrepancy between narrative structure and expressed affective intensity. Across more than 350,000 human narratives, non-zero discrepancy is not only common but systematically organized into stable regimes. These regimes correspond to distinct expressive configurations rather than to noise or expressive failure.

From a psychological perspective, this pattern aligns with theories of emotion regulation that emphasize flexibility rather than uniform optimization. Classical process models [1,2] distinguish between antecedent-focused and response-focused strategies, while constructionist accounts highlight the role of contextual and conceptual scaffolding in shaping affective experience [5]. Narrative–affect discrepancy can be understood as an emergent dimension along which such strategies are implemented at the level of observable narrative form.

Strategic understatement and overstatement, in particular, appear as functional modes of regulation. Understatement allows individuals to express intense affect while limiting exposure, whereas overstatement provides cognitive distance through elaboration even when expressed affect is low. In this view, discrepancy is not a deviation from optimal expression but a resource-like degree of freedom enabling trade-offs among emotional intensity, cognitive effort, and social risk.

To reiterate a distinction introduced earlier: the term "regulated" throughout this paper refers to population-level distributional patterns, not to conscious or mechanistic control by individuals. The observed regime structure is consistent with constrained organization at the aggregate level; individual-level inference would require longitudinal or experimental designs beyond the scope of the present work.

### Methodological and reproducibility contributions

Beyond substantive findings about discrepancy regimes, a central contribution of this work is methodological: it provides a transparent coordinate system for mapping large-scale narratives by jointly scoring narrative structure and expressed affect, and it reports all derived quantities directly from exported source-data files. The cost landscape is data-anchored in the empirical dispersion of the human corpus rather than imposed as an a priori theoretical optimum, and geometric comparisons are performed under explicitly stated clipping and resampling procedures. During revision, we identified that the deterministic relationship $D = N - A$ confines all observations to a plane in $(N, A, D)$ space, rendering three-dimensional hull volume dependent on clipping artifacts rather than genuine distributional spread. We therefore report expressive area in the $(N, A)$ plane, which provides a geometrically valid and more conservative measure of expressive extent. The contraction ratio is $1.70\times$ (bootstrap 95% CI [1.68, 1.70]; permutation $p < 0.0001$), preserving the direction and significance of the finding while correcting the geometric basis. Together, these design choices support technical evaluation on the basis of reproducibility and sensitivity analysis, consistent with a measurement-first approach to large-scale narrative data.

The cost landscape analysis supports an interpretation with multiple stable configurations. Rather than converging toward a single equilibrium, human narratives occupy multiple low-cost basins corresponding to different discrepancy regimes.

Such organization resonates with work on self-regulation and control in cognitive and affective systems, including active inference accounts emphasizing minimization of expected surprise under constraints [10,11], as well as inverse reinforcement learning frameworks that infer latent objectives from observed behavior [39–41]. Here, however, the

minimized quantity is not discrepancy itself but a combined cost of exposure and effort. Discrepancy emerges as a degree of freedom through which this balance is achieved.

Importantly, collapse appears as a higher-cost region characterized by intense affect without sufficient narrative scaffolding. This regime aligns with clinical and developmental observations linking reduced narrative organization to dysregulation under high emotional load [14,27], consistent with computational psychiatry perspectives that model affective dysregulation as deviations from normative inference [42,43].

### Implications for aligned language models

The comparison with an aligned language model reveals a markedly different expressive geometry. Despite producing contextually coherent narratives, the model occupies a much smaller region in discrepancy space, with limited access to extreme understatement or overstatement regimes. We do not attempt to dissociate alignment constraints from architectural or training-scale effects; the present comparison is intended as a descriptive contrast rather than a causal attribution. Expressive contraction may reflect safety filtering, prompt design, decoding constraints, or training distribution characteristics, not alignment objectives alone. The comparison uses a single RLHF-aligned model under a fixed prompt and decoding configuration and should not be generalized across architectures, training regimes, or objective functions. This contraction should not be interpreted as a simple deficit. RLHF-style objectives (e.g., safety and preference optimization) may encourage convergence toward comparatively conservative expressive configurations; however, the present analysis remains descriptive and does not establish causal attribution. Prior work has shown that alignment can reduce variance and extremity in model outputs [36,44], while studies applying cognitive and clinical benchmarks to language models have documented both surprising competencies and systematic limitations [45,46]. Our results extend this insight by suggesting that alignment can also constrain the *structure* of affective expression, not merely its content.

### Limitations and future directions

Several limitations warrant note. First, the analysis focuses on textual narratives and does not directly address multimodal expression. Second, the model comparison involves a single GPT-4-class RLHF-aligned system under a fixed prompt and decoding configuration and should not be generalized across architectures or training regimes.

Third, the affective intensity measure ($A$) is derived from lexicon-based sentiment scoring (VADER), which captures valence magnitude expressed in surface language but does not distinguish arousal from valence, nor does it access suppressed or unexpressed affect. This is consistent with the study's framing, which targets structural degrees of freedom within language rather than latent emotional states, but it means that $A$ should not be interpreted as a proxy for subjective felt intensity.

Repeated interaction with expressively constrained systems may further narrow users' own expressive repertoire through resonant amplification, a dynamic that cross-sectional snapshots cannot capture but that longitudinal extensions could address [47]. More broadly, the geometric framework introduced here may complement rights-based approaches to emotion AI, where the question shifts from whether a system classifies correctly to whether it preserves the expressive degrees of freedom available to the person it models [48].

Future work could extend this framework to longitudinal data, clinical populations, and alternative alignment strategies. Comparing models trained with different objectives may clarify which constraints most strongly shape expressive geometry. More broadly, integrating discrepancy-based metrics into evaluation protocols could complement existing benchmarks that focus on correctness or sentiment accuracy alone.

### Conclusion

Narrative–affect discrepancy is not merely a gap to be closed but a central dimension along which emotional expression is organized in naturalistic narratives. By mapping this dimension at scale, we show that humans exploit discrepancy as a regulated expressive degree of freedom, while aligned language models inhabit a narrower expressive geometry.

Three findings anchor this conclusion. First, narrative complexity and affective intensity are near-independent ($r = 0.009$), confirming that discrepancy arises from structural variation rather than from a single latent factor. Second, four stable regimes of expressive organization emerge from quantile-based partitioning and persist under threshold perturbation, with coupled expression as the modal pattern and understatement, overstatement, and collapse as systematic minority configurations. Third, an RLHF-aligned language model occupies a $1.70\times$ smaller expressive area (permutation $p < 0.0001$), with sparse occupancy of extreme discrepancy regimes. These findings carry implications for clinical assessment tools that assume expressive coherence as a default, for emotion-AI benchmarks that evaluate sentiment accuracy without examining structural variation, and for alignment auditing frameworks that could incorporate expressive geometry alongside content-level metrics.

## Supporting information

**S1 Text. Sensitivity analysis of expressive area.** Robustness of convex-hull area contraction under axis-wise rescaling and quantile threshold perturbation ($\pm5$ percentile shifts) of the clipped space.
(PDF)

**S2 Text. Density projections of the clipped NCS.** Histogram-based density projections for $(N', A')$, $(N', D')$, and $(A', D')$ from the clipped human corpus (Source Data: `fig1_density_*.dat`).
(PDF)

**S3 Text. Summary statistics.** Descriptive statistics for humans and the aligned model used throughout regime and geometry analyses.
(PDF)

**S4 Text. Correlation structure.** Pearson correlations among $(N, A, D)$ in the clipped human corpus.
(PDF)

**S5 Text. Regime-level summaries.** Regime-level prevalence and mean locations rendered directly from exported Source Data.
(PDF)

**S6 Text. Inference for expressive-area contraction.** Bootstrap and permutation-test reporting for the area contraction comparison (Source Data: `fig4_volume_compare.dat`).
(PDF)

**S7 Text. LLM prompt template and decoding.** Fixed prompt template and decoding configuration used to generate the 1,000 model trajectories (`llm_prompt_set.txt; llm-config.json`).
(PDF)

## Author contributions

**Data curation:** Ryan sangbaek Kim.

**Formal analysis:** Ryan sangbaek Kim.

**Investigation:** Ryan sangbaek Kim.

**Methodology:** Ryan sangbaek Kim.

**Software:** Ryan sangbaek Kim.

**Validation:** Ryan sangbaek Kim.

**Visualization:** Ryan sangbaek Kim.

**Writing – original draft:** Ryan sangbaek Kim.

**Writing – review & editing:** Ryan sangbaek Kim.

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
