## [Decision Letter · Decision Letter 0]

24 Mar 2026

PONE-D-26-02582Narrative–Affect Discrepancy as a Regulated Degree of Freedom in 351,734 Relationship NarrativesPLOS One

Dear Dr. Kim,

Thank you for submitting your manuscript to PLOS ONE. After careful consideration, we feel that it has merit but does not fully meet PLOS ONE’s publication criteria as it currently stands. Therefore, we invite you to submit a revised version of the manuscript that addresses the points raised during the review process.

We look forward to receiving your revised manuscript.

Kind regards,

Vanessa Carels

Staff Editor

PLOS One

Journal Requirements:

Reviewers' comments:

Reviewer's Responses to Questions

**Comments to the Author**

1. Is the manuscript technically sound, and do the data support the conclusions?

Reviewer #1: Yes

Reviewer #2: Yes

Reviewer #3: Yes

2. Has the statistical analysis been performed appropriately and rigorously? 

Reviewer #1: Yes

Reviewer #2: No

Reviewer #3: Yes

3. Have the authors made all data underlying the findings in their manuscript fully available?

Reviewer #1: Yes

Reviewer #2: Yes

Reviewer #3: Yes

4. Is the manuscript presented in an intelligible fashion and written in standard English?

Reviewer #1: Yes

Reviewer #2: Yes

Reviewer #3: Yes

5. Review Comments to the Author

Reviewer #1: Reviewer’s Comments to the Author

Clarity and Structure

The document is generally well organized, but some sections would benefit from clearer transitions. Consider adding brief linking sentences to improve coherence between paragraphs and ideas.

Content Depth and Critical Analysis

The topic is relevant and appropriately addressed; however, certain sections rely more on description than analysis. Strengthening critical reflection and supporting arguments with examples or evidence would enhance the overall quality.

Language and Expression

The language is mostly clear, but there are a few grammatical and syntactical errors. A careful proofreading is recommended to improve readability and maintain academic tone.

Consistency and Formatting

Please ensure consistency in formatting, headings, and terminology throughout the document. Minor inconsistencies affect the professional presentation of the work.

Alignment with Objectives/Guidelines

While the key objectives are addressed, explicitly linking reflections or findings to the stated learning outcomes or guidelines would make the work more focused and aligned with expectations.

Conclusion and Reflection

The conclusion could be strengthened by more clearly summarizing key insights and reflecting on their implications for future practice or learning.

Reviewer #2: The manuscript addresses an interesting and relevant research problem and presents a structured methodology for the proposed study.

Reviewer #3: The research is exceptionally well-structured and technically sound. The authors utilize a massive dataset of 351,734 naturalistic relationship narratives, providing a highly robust sample size. The mathematical formalization of the Narrative-Control Space (NCS) using narrative complexity (N), linguistically inferred affective intensity (A), and discrepancy (D) is logically sound and theoretically grounded. The comparative analysis mapping an RLHF-aligned Large Language Model (LLM) into the same coordinate system is an innovative and effective probe that strongly supports the conclusions regarding human expressive degrees of freedom and AI language models' contraction.

This is a highly compelling and original piece of research that successfully bridges computational social science, emotion regulation, and AI alignment. The geometric approach to mapping "expressive degrees of freedom" is a novel contribution that smartly moves beyond traditional sentiment analysis paradigms.

A minor suggestion: While you explicitly acknowledge early on that your focus is on the structural degrees of freedom within language rather than attempting to directly measure subjective, hidden feelings, it might be beneficial to briefly expand in the discussion on the limitations of the specific sentiment-derived measures (A) used.

Overall, the methodology is sound, the data transparency is exemplary, and the findings constitute a significant contribution to the field.

6. PLOS authors have the option to publish the peer review history of their article (what does this mean?). If published, this will include your full peer review and any attached files.

Reviewer #1: No

Reviewer #2: **Yes:** Mrinmoy Kayal

Reviewer #3: No

---

## [Author Response · Author response to Decision Letter 1]

27 Mar 2026

Please see the attached file "PONE-D-26-02582_R1_Response Letter.pdf" for our complete, formatted Response to Reviewers, including point-by-point replies to all reviewer and editor comments, a summary table of revisions, and an author-initiated correction note.

A brief summary of the key revisions:

1. Geometric correction: All expressive-extent measures recomputed as convex-hull area in the 2-D clipped (N', A') plane, replacing the 3-D hull volume reported in the original submission.

2. Updated statistics: Human hull area = 99.514 (bootstrap mean; 95% CI [98.596, 99.755]); LLM hull area = 58.679; contraction ratio = 1.696 (95% CI [1.680, 1.700]); permutation p < 0.0001 (T = 10,000); occupancy entropy: Human = 4.654, LLM = 5.103; Pearson r(N, A) = 0.009.

3. Terminology unified to Narrative-Control Space (NCS) throughout.

4. Cost function J justified as abductive and illustrative (Methods 2.6).

5. Regime thresholds clarified as quantile-based and data-driven; sensitivity analyses added (S1 Text).

6. "Collapse" label retained and justified as descriptive, non-clinical.

7. LLM comparison reframed as descriptive contrast; alternative explanations acknowledged.

8. VADER-based affect limitation added (Limitations 4.4).

9. Summary table of constructs and regimes added (Table 1).

10. Conclusion strengthened with three anchoring findings.

11. Code Availability statement added; Supplementary Software bundle included with standalone reproduction script (ncs_reproduce.py).

12. Data Availability updated with all-versions Zenodo DOI (10.5281/zenodo.17632585), v1.1.0 and v1.3.0 identified.

13. LLM model identifier specified: OpenAI gpt-4.1-mini, temperature 0.9, top-p 1.0, max tokens 600.

All revisions are detailed in the attached Response Letter with page references to the revised clean manuscript.

---

## [Decision Letter · Decision Letter 1]

20 Apr 2026

Narrative–Affect Discrepancy as a Regulated Degree of Freedom in 351,734 Relationship Narratives

PONE-D-26-02582R1

Dear Dr. Kim,

We’re pleased to inform you that your manuscript has been judged scientifically suitable for publication and will be formally accepted for publication once it meets all outstanding technical requirements.

Kind regards,

Gea Oliveri Conti, Ph.D. MBs

Academic Editor

PLOS One

Additional Editor Comments (optional):

Reviewers' comments:

Reviewer's Responses to Questions

**Comments to the Author**

1. If the authors have adequately addressed your comments raised in a previous round of review and you feel that this manuscript is now acceptable for publication, you may indicate that here to bypass the “Comments to the Author” section, enter your conflict of interest statement in the “Confidential to Editor” section, and submit your "Accept" recommendation.

Reviewer #1: All comments have been addressed

2. Is the manuscript technically sound, and do the data support the conclusions?

Reviewer #1: Yes

3. Has the statistical analysis been performed appropriately and rigorously? 

Reviewer #1: Yes

4. Have the authors made all data underlying the findings in their manuscript fully available?

Reviewer #1: Yes

5. Is the manuscript presented in an intelligible fashion and written in standard English?

Reviewer #1: Yes

6. Review Comments to the Author

Reviewer #1: This manuscript presents a novel and ambitious large-scale analysis of narrative–affect discrepancy using an exceptionally large dataset of 351,734 relationship narratives. The central contribution—conceptualizing discrepancy (D = N − A) as a structured and regulated dimension rather than noise—is both original and potentially impactful across computational social science, narrative psychology, and affective computing. The introduction is well-motivated, clearly situating the work against prevailing assumptions that coherence between affect and narrative is normative. The shift toward a geometric and population-level framing is particularly compelling.

The methodology is generally rigorous and transparent. The use of a two-dimensional expressive space (N, A) with derived discrepancy is appropriate, and the correction from 3D volume to 2D convex hull area strengthens the validity of the geometric analysis. The inclusion of clipping, bootstrapping, and permutation testing enhances robustness, and the effort toward reproducibility—through shared datasets, code, and detailed procedural descriptions—is commendable. The regime classification is intuitive and empirically grounded, and the sensitivity analyses increase confidence in the stability of findings.

However, there are several areas that would benefit from clarification or further justification. First, the operationalization of affective intensity using VADER, while practical, remains a limitation. Although acknowledged, the manuscript could more explicitly discuss how this choice may bias the results, particularly in detecting subtle or context-dependent emotional expression. Second, the interpretation of “regulation” at the population level should be handled with continued caution to avoid overextension into psychological claims about individual behavior. While the authors provide this caveat, it would be helpful to reinforce it in the discussion of implications.

The comparison with the RLHF-aligned language model is interesting but should remain clearly framed as exploratory. The manuscript appropriately avoids strong causal claims; however, additional discussion of alternative explanations (e.g., prompt design, decoding constraints, or dataset mismatch) would strengthen this section. Including comparisons with multiple models or configurations in future work could address this limitation.

From an ethics perspective, the study appears compliant with data privacy norms, given the use of de-identified, derived features and publicly उपलब्ध data. There are no immediate concerns regarding research ethics, dual publication, or conflicts of interest. The data availability and transparency statements are strong and align with open science practices.

In conclusion, this is a thoughtful and methodologically careful contribution that introduces a valuable framework for analyzing emotional expression in narratives. With minor clarifications and expanded discussion of limitations, the manuscript is suitable for publication and likely to stimulate further research in this area.

7. PLOS authors have the option to publish the peer review history of their article (what does this mean?). If published, this will include your full peer review and any attached files.

Reviewer #1: **Yes:** Dr Abhipriya Roy

---

## [Editor Report · Acceptance letter]

PONE-D-26-02582R1

PLOS One

Dear Dr. Kim,

I'm pleased to inform you that your manuscript has been deemed suitable for publication in PLOS One. Congratulations! Your manuscript is now being handed over to our production team.

Kind regards,

on behalf of

Dr. Gea Oliveri Conti

Academic Editor

PLOS One